# Prognostic Impact of Tumor-Infiltrating Lymphocytes and Neutrophils on Survival of Patients with Upfront Resection of Pancreatic Cancer

**DOI:** 10.3390/cancers11010039

**Published:** 2019-01-03

**Authors:** Rainer C. Miksch, Markus B. Schoenberg, Maximilian Weniger, Florian Bösch, Steffen Ormanns, Barbara Mayer, Jens Werner, Alexandr V. Bazhin, Jan G. D’Haese

**Affiliations:** 1Department of General, Visceral, and Transplantation Surgery, Ludwig-Maximilians-University Munich, Marchioninistr. 15, 81377 Munich, Germany; Rainer.Miksch@med.uni-muenchen.de (R.C.M.); markus.schoenberg@med.uni-muenchen.de (M.B.S.); maximilian.weniger@med.uni-muenchen.de (M.W.); Florian.Boesch@med.uni-muenchen.de (F.B.); Barbara.Mayer@med.uni-muenchen.de (B.M.); jens.werner@med.uni-muenchen.de (J.W.); Jan.DHaese@med.uni-muenchen.de (J.G.D.); 2Department of Pathology, Ludwig-Maximilians-University Munich, Marchioninistr. 15, 81377 Munich, Germany; Steffen.Ormanns@med.uni-muenchen.de; 3German Cancer Consortium (DKTK), Partner Site Munich, Pettenkoferstr. 8a, 80336 Munich, Germany

**Keywords:** tumor-infiltrating lymphocytes, tumor microenvironment, immunoscore, immune cell infiltration, immune infiltrate, immunosuppression, quantification of the tumor immune stroma (QTiS), activated stroma index, pancreatic ductal adenocarcinoma

## Abstract

In patients with pancreatic ductal adenocarcinoma (PDAC), the tumor microenvironment consists of cellular and stromal components that influence prognosis. Hence, tumor-infiltrating lymphocytes (TILs) may predict prognosis more precisely than conventional staging systems. Studies on the impact of TILs are heterogeneous and further research is needed. Therefore, this study aims to point out the importance of peritumoral TILs, tumor-infiltrating neutrophils (TINs), and immune subtype classification in PDAC. Material from 57 patients was analyzed with immunohistochemistry performed for CD3, CD8, CD20, CD66b, α-sma, and collagen. Hot spots with peritumoral TILs and TINs were quantified according to the QTiS algorithm and the distance of TILs hot spots to the tumor front was measured. Results were correlated with overall (OS) and progression-free survival (PFS). High densities of peritumoral hot spots with CD3^+^, CD8^+^, and CD20^+^ TILs correlated significantly with improved OS and PFS. Combined immune cell subtypes predicted improved OS and PFS. High infiltration of CD3^+^ TILs predicted progression after 12 months. The location of TILs’ hot spots and their distance to the tumor front did not correlate with patient survival. Peritumoral TILs and the composition of the stroma predict OS and PFS in PDAC.

## 1. Introduction

Pancreatic ductal adenocarcinoma (PDAC) ranks fourth amongst cancer-associated deaths in both men and women [1,2,3,4,5,6]. Despite constant efforts to improve treatment, the prognosis remains poor, with an overall five-year survival rate of about 5% [5,7]. This can be attributed to lack of efficient chemo- and radiotherapy options, but more importantly, to the presentation of patients mostly in advanced tumor stages [8,9].

Histologically, PDAC is described primarily by the transformation of exocrine cells to tumor cells [10]. Here, the focus is on cell differentiation and migration, which is also represented by the TNM Classification of Malignant Tumours (TNM). Besides features of PDAC cells themselves, the complex function of the tumor microenvironment (TME) has become increasingly important [11] in explaining development, growth, and metastasis of PDAC [12]. The anti-cancer immune response, with its different immune cells and their interaction with the pancreatic tumor cells, has an influence on carcinogenesis and progression as well as on survival of cancer patients [11], also in the case of PDAC [13,14].

Growing evidence suggests that the TME, and especially accumulation of tumor-infiltrating leukocytes, mostly lymphocytes (TILs) [11,12,15,16], is important for prognosis in PDAC [14,17,18,19]. Quantification of TILs has been shown to predict survival more precisely than the TNM classification system in colorectal cancer but its prognostic power in cases of PDAC is subject of an ongoing debate [20,21,22,23]. Thus, TIL scoring systems may bear the potential to predict prognosis in PDAC [22]. While conventional tumor classification systems rely on only tumor size or infiltration, lymph node metastasis, and distant metastasis, immunological scoring systems represent an important tool for characterizing the tumor microenvironment [24]. As shown in breast [25,26], lung [27], and colorectal cancer [23], immunoscoring systems help to predict prognosis more precisely than the TNM classification by subtype classification. Furthermore, classification of subtypes in PDAC provides a guide for clinical decisions to adjust treatment. This has already been shown in genetic and molecular quantifications in patients with PDAC [28,29], where high expression levels of GATA-binding protein 6 and of genes associated with adhesion, epithelium were associated with prolonged survival as compared to low expression. GATA 6 is a marker of response to adjuvant chemotherapy [30]. However, therapeutic approaches focusing on targets definded by genomics may only be relevant for a small proportion of patients with PDAC [31]. Thus, in comparison to conventional TNM staging and genomic approaches, PDAC could be characterized more accurately by quantifying tumor cells in combination with analyzing the corresponding TME [20]. 

The TME in PDAC is complex and consists of cellular and stromal components. With respect to the cellular part, PDAC stroma is dominated by cancer-associated fibroblasts (CAFs), which interact with PDAC cells and have a crucial influence on patient survival [32,33]. More precisely, by producing abundant amounts of peritumoral stroma, CAFs reduce PDAC perfusion and are thought to be responsible for the profound chemoresistance found in PDAC [34].

Interestingly, the TME of PDAC is immunosuppressive by nature due to a variety of mechanisms helping PDAC cells to escape the anti-tumor immune response consisting of tumor-infiltrating leukocytes, especially TILs, TINs, M2 macrophages, and myeloid-derived suppressor cells [35]. As another point corroborating the immunosuppressive phenotype of PDAC, the efficacy of immunotherapeutic approaches such as checkpoint inhibitors is limited to small subsets of PDAC patients [36,37,38]. Thus, tumor-infiltrating immune cells are of particular interest for both quantification of response to therapy and subclassification of PDAC [11].

With the present study, we are aiming to show the influence of peritumoral tumor-infiltrating leukocytes with TILs and TINs on survival of patients with upfront resection of PDAC and its correlation with stromal composition.

## 2. Results

### 2.1. Demographical Data, Tumor Size, and Grade of Necrosis

Fifty seven (57) patients who underwent upfront resection for PDAC were included in this study. The mean age was 70.4, with 54.4% of patients being female. None of the patients underwent preoperative chemotherapy. 23 pylorus-resecting pancreaticoduodenectomies (PrPD), 15 pylorus-preserving pancreaticoduodenectomies (PPPD), 13 panreatic tail resections, and six total pancreatectomies were performed (Table 1). The median progression-free survival (PFS) was 19.0 months (standard deviation (SD) ± 10.185), and median overall survival (OS) was 19.0 months (SD ± 15.239). Fifty six (56) patients had tumor recurrence or metastasis during follow-up, and 12 patients (21.1%) were alive at the end of the follow-up period (with a median follow-up of 19.0 months). Sixteen patients (28.1%) completed adjuvant chemotherapy with gemcitabine, whereas 41 patients (71.9%) did not complete gemcitabine treatment due to death or progression.

Presence of extensive necrosis was related to impaired overall survival (*p* < 0.001) and progression-free survival (*p* = 0.009) (Appendix A). Bigger tumor size was related with extensive grade of necrosis in the regression analysis (*p* = 0.000526). Whereas, tumor size did not correlate with overall/progression-free survival, tumor-infiltrating leukocytes, and the tumor stroma significantly (Table 2 and Table 3).

### 2.2. Accumulation of Peritumoral TILs in PDAC Correlates with Better Survival

Hot spots of TILs and TINs were observed predominantly in the peritumoral area. However, they did not pass the tumor front. Therefore, accumulation of tumor-infiltrating leukocytes was demonstrated for peritumoral hot spots and groups of these leukocytes were divided by high vs. low infiltration based on the median values (Table 1). The parameters from Table 1 were examined for statistical correlation with type of infiltration (Table 2). Pathological quantification of the TNM classification was performed postoperatively.

High infiltration of CD3^+^, CD8^+^, and CD20^+^ TILs, but not CD66b^+^ TINs, was related to better overall (CD3 *p* = 0.003, CD8 *p* = 0.010, CD20 *p* = 0.038, and CD66b *p* = 0.265) and progression-free survival (CD3 *p* = 0.008, CD8 *p* = 0.021, CD20 *p* = 0.020, and CD66b *p* = 0.289) in the Kaplan-Meier analysis (Figure 1).

Additionally, these four different immune cells were combined (Figure 2 and Figure 3). Significant results were retrieved for improved OS and PFS with combination of high infiltration with CD3^+^/CD8^+^ (OS *p* = 0.005, PFS *p* = 0.038), CD8^+^/CD20^+^ (OS *p* = 0.038, PFS *p* = 0.029), and CD20^+^/CD66b^+^ tumor-infiltrating leukocytes (OS *p* = 0.010, PFS *p* = 0.028).

Finally, groups of three and four different immune cell markers were analyzed: both high infiltration of CD3^+^/CD8^+^/CD20^+^ TILs (OS *p* = 0.002, PFS *p* = 0.0003) and high infiltration of CD3^+^/CD8^+^/CD20^+^/CD66b^+^ tumor-infiltrating leukocytes (OS *p* = 0.009, PFS *p* = 0.001) correlated significantly with improved survival in the Kaplan-Meier analysis (Figure 2 and Figure 3). 

Next, we addressed whether both—the immunological stromal composition represented by tumor-infiltrating leukocytes (TILs and TINs) and the stromal composition represented by collagen and CAFs had to be considered to predict survival in patients with PDAC: The density of stromal composition was different among the patients in this study: high density of activated fibroblasts (CAF) was associated with a trend to impaired OS. High expression of collagen in selected areas was related to improved OS (*p* = 0.017) and PFS (*p* = 0.009). Both parameters combined as shown above retrieved a subtype classification of the activated tumor stroma index. Here, results of the activated stroma index showed a positive influence on survival for dormant stroma types as opposed to dominantly fibrolytic stroma [32,33] (Appendix A).

### 2.3. CD3^+^, CD8^+^, and CD20^+^ TILs Predict Survival of Patients with PDAC after 24 Months

The area under the curve (AUC) was highest for CD20^+^ TILs (AUC = 0.729) and the combinations with high infiltration of CD3^+^, CD8^+^, and CD20^+^ TILs (Table 3, Appendix A). Groups with CD3, CD8, or CD20 antigens on the immune cells correlated significantly with improved survival 24 months after resection (*p* < 0.05).

### 2.4. Distance of TILs’ Hot Spots to the Tumor Front Has No Influence on Survival

Peritumoral infiltration of immune cells was mainly observed at the edge of the tumor front with no intratumoral hot spots at all (Appendix A). Therefore, we aimed to understand whether the distance between these peritumoral hot spots and the tumor front of PDAC were linked to survival and the stromal composition.

The dominant peritumoral hot spots of CD3^+^, CD8^+^ and CD20^+^ TILs were chosen because significant results were obtained in the survival analysis. Both cell types gave positive results regarding influence on survival and staining was declared as specific for a certain immune cell type. The Kaplan-Meier analysis did not reveal significant results concerning distance between immune cell hot spot and tumor front (Figure 4): *p*-values were as follows for OS and CD3^+^ hot spot-tumor front *p* = 0.723, CD8 *p* = 0.753, and CD20 *p* = 0.140; for PFS and CD3^+^ hot spot-tumor front *p* = 0.148, CD8 *p* = 0.723, and CD20 *p* = 0.901. Furthermore, the distance was not dependent on the quality of the tumor stroma (CAFs, collagen, active stroma index). The results suggested that the distance between hot spots and the tumor front did not have an influence on survival. 

### 2.5. High Accumulation of CD3^+^ TILs and High Density of Collagen Correlate with Improved Survival in a Multivariate Analysis

The dependent variables were analyzed in a univariate analysis (Table 4). Variables were then included in the multivariate analysis if they had retrieved *p*-values < 0.05 (Table 5) using a linear regression model with overall and progression-free survival as outcome parameters. Combined groups of several tumor-infiltrating leukocytes were not included in the multivariate analysis to avoid bias by double use of TILs as parameter in the linear regression analysis.

Ultimately, high infiltration of CD3^+^ TILs (*p* = 0.031) and high density of collagen (*p* = 0.014) correlated significantly with better overall survival. The same correlation was observed for progression-free survival: *p* = 0.021 for CD3^+^ TILs and *p* = 0.018 for collagen (Table 5).

## 3. Discussion

While treatment of other cancers has improved in the last decades, patients with PDAC still face a dire prognosis. Thus, enormous efforts need to be undertaken to improve both treatment and prognosis of PDAC. Presently, the understanding of the TME in PDAC is a field of growing interest [13,14,19,37,38]. Here, tumor-infiltrating lymphocytes are an important prognostic factor in different tumor entities [11,40,41]. Moreover, subtype classification of the TME may help to predict the efficacy of adjuvant therapies. 

In PDAC, clinical efforts to utilize immune therapy have been shown to be largely ineffective [42]. While patients with other cancers benefit from increased survival rates with immunological check-point inhibitors enhancing anti-tumoral immune cell activity, only small subsets of PDAC patients respond to these treatments [43,44,45]. Here, the dense fibrotic PDAC tumor stroma and the immunosuppressive microenvironment impede anti-tumoral immune response, making immune therapy of PDAC extraordinarily challenging [46]. Notably, pancreatic stellate cells have been shown to inhibit cytotoxic T cell infiltration through NFκB activation [46]. In comparison to these data implicating an immunosuppressive phenotype of PDAC, tumor infiltration with CD8^+^ TILs has been identified as an important factor determining long-term survival in PDAC [47].

Therefore, this study aimed to determine the importance of peritumoral tumor-infiltrating leukocytes (CD3^+^, CD8^+^, CD20^+^ TILs, and CD66b^+^ TINs and their combinations) in stratifying patient prognosis. Furthermore, quantification of TILs and TINs may be a possible tool for additional immunological subtype classification of patients with PDAC [48,49].

We showed that high infiltration of CD3^+^, CD8^+^, and CD20^+^ TILs correlates with improved overall and progression-free survival. Strikingly, infiltration with CD3^+^ TILs is a factor predicting survival independent of classical histopathological parameters. Thus, these data imply that the immunological anti-tumor response needs to be considered when predicting patient prognosis and outcome.

The prognostic importance of TILs were described in patients with PDAC previously [12,50,51,52,53,54,55]. However, the methods and location of the described immune cells differ considerably across the literature [40]. With respect to the location of immune cells, we only observed single intratumoral cells but no hot spots, which might be explained by the ability of PDAC cells to escape immune response [11,56,57]. On the one hand, many studies use tumor microarrays (TMAs) as material [20,54,55,58]. Whereas, on the other hand, there are studies in which immunohistochemistry is still performed on whole-slide paraffin sections [12,50,52,59,60].

While TMAs are more efficient and space-saving than whole-slide tissues, they provide less information on cellular information as compared to whole-slide imaging. With respect to the TME, whole-slide tissues are more similar to the original tumor [61,62,63,64]. Therefore, bigger slides provide an opportunity to have a more differentiated view of the tumor tissue.

Nevertheless, the understanding of immune cells localization in PDAC is not exhaustive. The analysis of the observed immune cells are heterogenous: specific areas or hot spots are observed, TMA or whole slide tissue is used, and peritumoral or intratumoral areas are investigated [20]. In the present study we focused on the peritumoral hot spots which were located at the tumor front. This tumor front was defined as invasive margin where the peritumoral hot spots were located and did not pass. Therefore, the leukocytes were not able to infiltrate as intratumoral hot spots. These predominantly peritumoral hot spots in the present study supported the hypothesis of a tumor front in PDAC [57] which represents an escape mechanism against the immune response. In particular, cytotoxic cells are not able to attack tumor cells [65]. Thus, mechanisms to overcome this problem in PDAC are still investigated. Unfortunately, immunomodulating therapy in PDAC did not show the same hopeful results like in other tumor entities.

Furthermore, the present study aimed to address the question whether selected areas [12,52,54,55,58,60] or hot spots [50,59]—high densities of certain immune cells—should be used to quantify the immune cells. Some authors emphasize the accumulation of the location of TILs. It therefore seemed promising to analyze hot spots rather than single scattered cells.

We compared the importance of peritumoral CD3^+^ hot spots and intratumoral selected areas of CD3^+^ cells. Here, high infiltration of CD3^+^ TILs retrieved most promising results regarding prediction of prognosis: intratumoral CD3^+^ TILs correlated with improved PFS (*p* = 0.047) as well as peritumoral CD3^+^ hot spots (*p* = 0.008), but did not retrieve accurate results for OS in the Kaplan-Meier analysis (*p* = 0.142) compared to the significant correlation of peritumoral CD3^+^ hot spots on OS (*p* = 0.003). If CD3^+^ TILs infiltration was high in the peritumoral hot spot, the intratumoral infiltration of CD3^+^ TILs had been increased as well (*p* = 0.007). The next question was wether the distance to the tumor front influenced the intratumoral infiltration: there was no significant correlation.

Tahkola et al. observed CD3^+^ and CD8^+^ TILs in the tumor core and invasive margin with similar results as compared to the present study [20]. However, intratumoral hot spots as described in the tumor core by Tahkola et al. or as by Galon et al. [24] were not detected. Ultimately, this was the reason for our hypothesis that greater proximity of hot spots to the tumor front is related to worse survival. There was no significant correlation but a trend was observed towards CD20^+^ TILs for OS and CD3^+^ TILs for PFS. Finally, it is showing a possible importance in the antitumor immune response in PDAC. The location instead of the distance to the tumor front may be a possible tool to describe the importance of TIL accumulation in the TME. However, there was no correlation between the immune-tumor cell distance and patient survival in PDAC with this method. Furthermore, the immune escape mechanism and the antitumor response cannot be quantified by distance measurement with our method.

The immunosuppression in PDAC within the tumor microenvironment is thought to impair the host’s antitumor response by multiple immunosuppressive cells like myeloid-derived suppressor cells (MDSC) [66]. MDSCs can suppress T cell effector cells and induce immune tolerance for immune evasion [67]. A complex system of T cells, pancreatic tumor cells, myeloid-derived suppressor cells, tumor-associated macrophages, regulatory T cells, and pancreatic stellate cells (PSCs) seem to interact. There is still more evidence and research needed to totally understand the complex tumor microenvironment [68].

Finally, we showed that the QTiS algorithm for different tumor types described previously [39,69] retrieved relevant results to highlight the importance of peritumoral TILs in patients with PDAC. Results with colorectal cancer indicate that quantification of immune cell infiltration has the potential to become an integral part of the TNM classification of PDAC [70].

## 4. Material and Methods

### 4.1. Patients and Clinical Data

All experimental procedures were approved by the Human Tissue and Cell Research (HTCR) Foundation (2016-04) and the Ethics Committee of the University of Munich (807-16). Tumor samples and clinical data were collected from 65 patients. Material preparation and immunohistochemistry were performed independently from the collection of patients’ clinical data. In addition to demographical data, serological markers, and survival was recorded. Patients were excluded if survival data was unobtainable, patients were lost to follow-up or clinical data was incomplete. Ultimately, 57 patients who underwent upfront resection in between January 2014 and December 2015 were included in the analysis.

### 4.2. Material

Four-µm sections of paraffin slides obtained from patients with PDAC were provided from the pathology department. Primary antibodies—anti-CD3 (ab5690), anti-CD8 (ab17147), anti-CD20 (ab78237), anti-CD66b (ab197678), and anti-α-sma (ab5694)—were purchased from Abcam PLC (Cambridge, UK). A Masson-Goldner staining kit (100485) for collagen staining was bought from Merck KGaA (Darmstadt, Germany). Secondary antibodies—horse anti-mouse IgG (BA-2000) and horse anti-rabbit IgG (BA-1100)—were purchased from VECTOR Laboratories (Burlingame, CA, USA), along with the Avadin/Biotin Blocking Kit (SP-2001), the alkaline phosphatase enzyme detection system (ABC-AP) (AK-5000), and the ImmPACT Vector Red Alkaline Phosphatase Substrate Kit (SK-5105). 

### 4.3. Immunohistochemistry

Immunohistochemistry (IHC) for CD3 (T cell marker), CD8 (marker of cytotoxic T cells), CD20 (B cell marker), CD66b (marker of neutrophils), and α-sma (marcer of cancer associated fibroblasts) was performed as described previously using paraffin slides of 4-µm sections [39]: first, the paraffin slides were dewaxed and hydrated with xylene, ethanol, and distilled water. Second, antigens were retrieved with citrate buffer (pH 6.0 for antibodies to CD3, CD20, CD66b, and α-sma) or EDTA solution (pH 8.0 for antibody to CD8). Incubation of primary (working concentration: anti-CD3 1:50, anti-CD8 1:50, anti-CD20 1:400, anti-CD66b 1:50, and anti-α-sma 1:200) and secondary antibodies (working concentration: anti-mouse 1:200 for CD8 and anti-α-sma; anti-rabbit 1:200 for CD3, CD20, and CD66b) was performed over night at 4 °C. Alkaline phosphatase was used for immunostaining with the ABC-AP-kit for 30 min at room temperature. Counterstating was carried out with hematoxylin. Additionally, negative controls were implemented. IHC for collagen was performed with collagen-specific aniline blue from the Masson trichrome staining kit [33].

### 4.4. Analysis of Immunohistochemistry

First, areas with the highest infiltration density of CD3^+^, CD8^+^, CD20^+^, or CD66b^+^ cells were defined as hot spots [50,59] (Figure 5). This hot spot represents an accumulation of a certain immune cell type compared to single and spread immune cells. In addition, single immune cells were observed.

Quantification of the positive cells was performed using the subjective threshold of ImageJ software (Version 1.51h, National Institutes of Health, Bethesda, MD, USA). Under the microscope a tumor front which marked the edge of the peritumoral hot spots was observed. This tumor front was defined as margin where peritumoral hot spots were located and did not pass this frontier. Where feasible, 3 hot spots per antigen were quantified and median values calculated (Table 1). Single cells were distributed over the tissue, whereas no hot spots of intratumoral infiltrating cells were observed. TILs and TINs were counted in representative areas according to most publications—especially intratumoral—than selected hot spots with highest infiltration density in a review about TILs in different tumor entities [40].

The analysis of IHC for α-sma and collagen was performed in a modified procedure as compared to the protocol published by Erkan et al. [32,33]: three spots per slide per staining were selected and density was quantified using the subjective threshold of ImageJ software. The corresponding areas for α-sma and collagen were selected. The location of CD3^+^, CD8^+^ and CD20^+^ TIL hot spots in relation to the tumor front was observed in a HPF (400× magnification) and microphotographs were taken. The biggest hot spot on each slide was selected and three different distances in mm were calculated by marking circles around the hot spot and the tumor front using ZEN 2 lite software (ZEN Version 2.0, Carl Zeiss, Oberkochen, Germany) (Appendix A). In a further step the distance class was divided in “close” and “far” by calculating the median value. Quality control of IHC was performed according to Maxwell et al. [71].

The activated stroma index firstly introduced by Erkan et al. [33] determines whether the tumor stroma is fibrolytic (α-sma value higher than the median value of the mean values and collagen value lower than the median value of the mean values), fibrogenic, inert, or dormant (in the low group for α-sma and in the high group for collagen). These four groups were formed by separation in quartiles (Appendix A). Additionally, α-sma-, and collagen-stained areas were compared with survival separately and independently of the activated stroma index.

The extend of necrosis within the tumor tissue was assessed by a board certified pathologist (SO) an haemotoxylin and eosin (HE) stained slide of the same sample used in the immunhistochemical stainings. Briefly, absent or insignificant necrosis of single cells was graded as score 0, spot like necrosis in limited zones within the tissue as score 1 and extensive or confluent necrotic areas within the tumor tissue as score 2. The largest tumor diameter assessed during gross sectioning on routine diagnostic workup was retrieved from the original pathology reports.

### 4.5. Statistical Analysis

Statistical analysis was performed using SPSS statistics software (Version 24.0, IBM, Armonk, NY, USA). *p*-values < 0.05 were defined for inclusion of variables in the multivariate analysis after univariate analysis. The alpha-level was set to 0.05. First, parameters were evaluated to exclude their statistical dependence on each other in a binary logistic regression model. The statistical tests used were linear regression with overall and progression-free survival as dependent variables, the Kaplan-Meier survival analysis for overall and progression-free survival, and binary linear regression with the activated stroma index as dependent variables. The independent variables were obtained from the patients and clinical data. Statistical analysis of the immunohistochemistry results was performed after classifying the patients into two groups, high and low, with respect to tumor-infiltrating leukocytes. Median values were used to classify the two groups. First the mean value was calculated for each antibody and each patient, and then the median value was identified from among the 57 different mean values. Groups were performed as pair, trio, and group of four of the CD3^+^, CD8^+^, CD20^+^, and CD66b^+^ tumor-infiltrating leukocytes if the patient had been included in the same type of infiltration—high or low for different cell types (Figure 2 and Figure 3): i.e., patients with both high infiltration of CD3^+^ and CD8^+^ TILs were selected for the high infiltration combined group (written as CD3 + CD8 in the figures/tables and CD3^+^/CD8^+^ in this text), whereas patients with low infiltration of at least one of these immune cells were introduced in the combined low infiltration group. Accordingly, Kaplan-Meier analyses were performed for every immune cell subtype and their combinations.

Receiver operating characteristic (ROC) curves were plotted. The graphs show specificity and sensitivity regarding the influence of the different TILs/TINs on overall and progression-free survival. The results regarding the prognostic relevance of tumor-infiltrating leukocytes on overall and progression free survival can be quantified by specificity and sensitivity in the receiver operating characteristic (ROC) curves for CD3^+^, CD8^+^, CD20^+^, and CD66b^+^ TILs and its combinations.

Statistical power analysis was performed with GPower (Version 3.1.9.3, Heinrich Heine Universität Düsseldorf, Düsseldorf, Germany) [72,73]: Post hoc power analysis retrieved a statistical power of 0.9814151 for the total sample size of 57.

## 5. Conclusions

In conclusion, the cellular and the stromal composition of the tumor microenvironment influences survival in PDAC patients. High infiltration of peritumoral CD3^+^, CD8^+^, and CD20^+^ TILs correlates with improved PFS and OS in PDAC in the univariate analysis. These findings highlight the importance of subtype classification in PDAC to predict prognosis precisely.

## Figures and Tables

**Figure 1 cancers-11-00039-f001:**
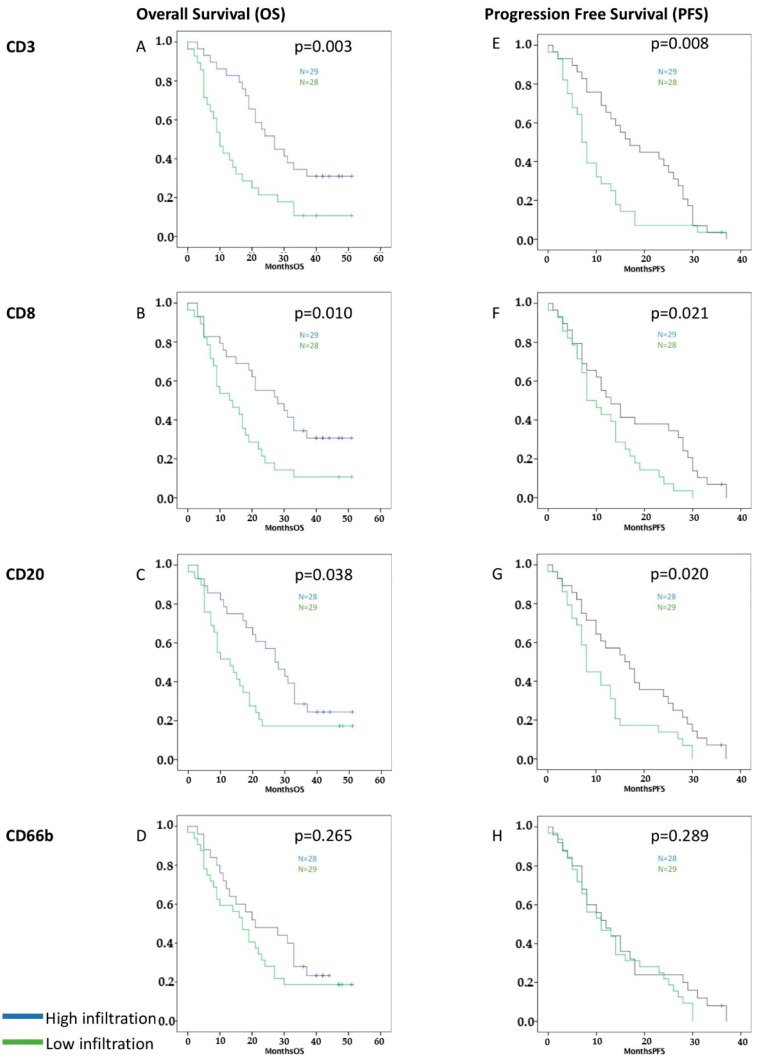
Survival graphs of overall (**A**–**D**) and progression-free survival (**E**–**H**) for high (blue) vs. low (green) infiltration of CD3^+^, CD8^+^, CD20^+^, and CD66b^+^ tumor-infiltrating leukocytes (TILs). The median value was used to distinguish high vs. low infiltration after quantification of stained cells [39]. High infiltration with CD3^+^, CD8^+^, and CD20^+^ TILs correlated significantly with improved OS and PFS. No statistically significant influence was evident for CD66b^+^ TINs.

**Figure 2 cancers-11-00039-f002:**
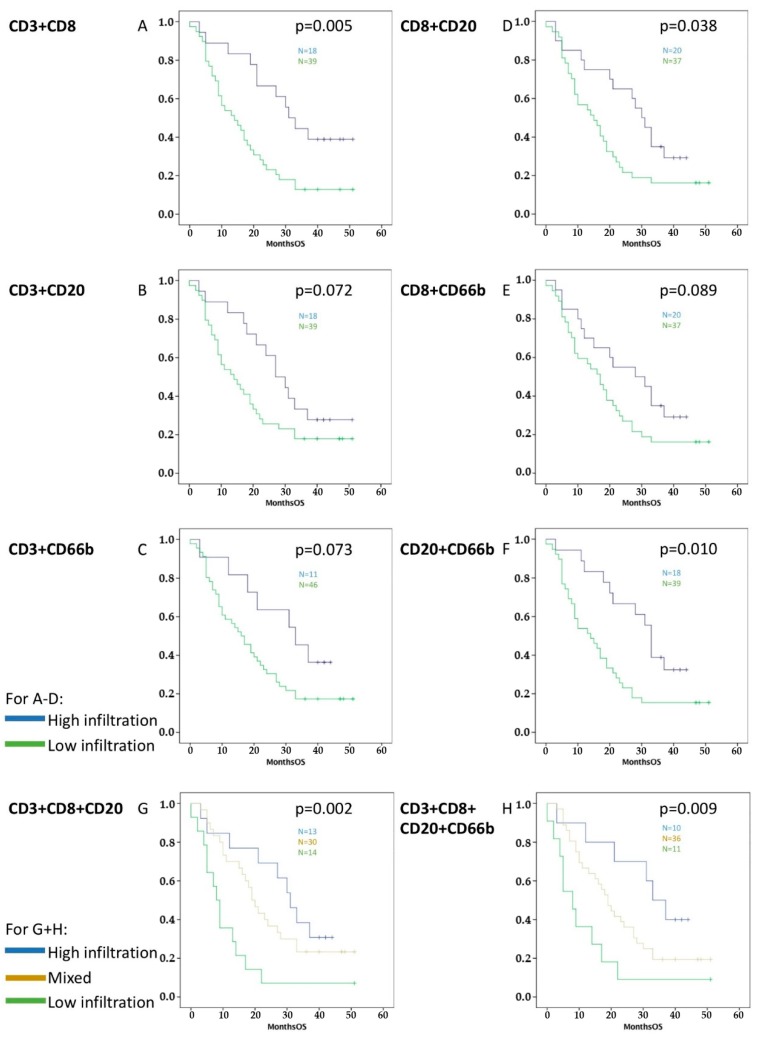
Survival graphs of OS in relation to different combinations of TILs and TINs: Groups were formed by separating high infiltration in both cell types (blue color, e.g., high CD3^+^ and high CD8^+^) from the rest (green color, e.g., high/low CD3^+^/CD8^+^ or low CD3^+^ and low CD8^+^), where two immune cells were analyzed together (**A**–**F**). Groups were formed by separating high infiltration in all cell types, low infiltration in all cell types, and mixed levels of infiltration, where three immune cells were analyzed together (**G**). Finally, combination with four immune cells was performed (**H**). High infiltration of CD3^+^/CD8^+^, CD3^+^/CD8^+^/CD20^+^, CD8^+^/CD20^+^, CD20^+^/CD66b^+^, and CD3^+^/CD8^+^/CD20^+^/CD20^+^/CD66b^+^ tumor-infiltrating leukocytes correlated significantly with better OS.

**Figure 3 cancers-11-00039-f003:**
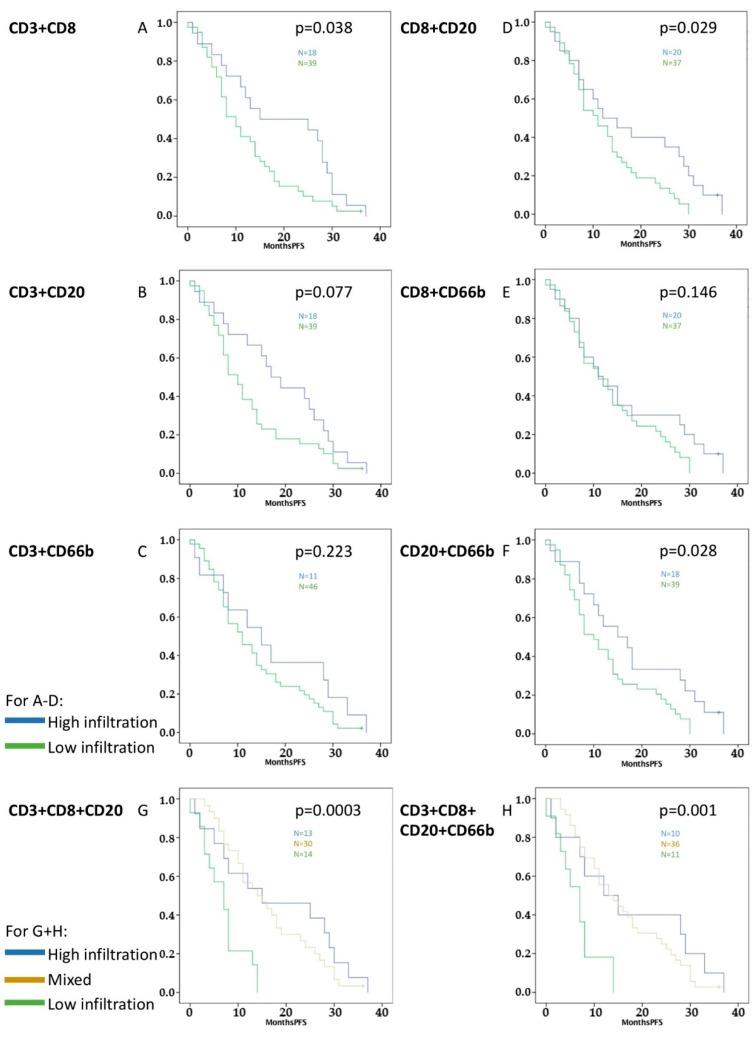
Survival graphs of PFS in relation to different combinations of TILs and TINs: Groups were formed by separating high infiltration in both cell types accordingly to Figure 2. (**A**–**F**) Groups were combined with two cell types, (**G**) with three, and (**H**) with four different cell types. High infiltration of CD3^+^/CD8^+^, CD3^+^/CD8^+^/CD20^+^, CD8^+^/CD20^+^, CD20^+^/CD66b^+^, and CD3^+^/CD8^+^/CD20^+^/CD20^+^/CD66b^+^ tumor-infiltrating leukocytes correlated significantly with better PFS.

**Figure 4 cancers-11-00039-f004:**
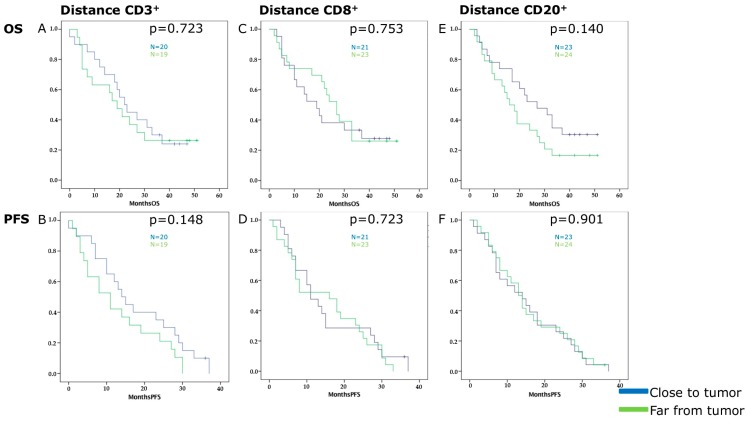
Kaplan-Meier survival curves of overall (**A**,**C**,**E**) and progression-free survival (**B**,**D**,**F**) in relation to the distance in µm of CD3^+^ (**A**,**B**), CD8^+^ (**C**,**D**) or CD20^+^ (**E**,**F**) hot spots from the tumor front. The groups are divided into hot spots close to the tumor front (blue) and hot spots far from tumor front (green) according to the median distance value: 0.0333253 µm for CD3, 0.0687577 µm for CD8, and 0.056765 µm for CD20.

**Figure 5 cancers-11-00039-f005:**
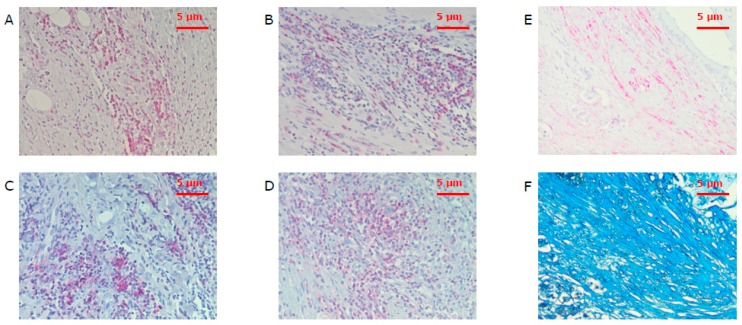
Hot spots with tumor-infiltrating leukocytes of CD3^+^ (**A**), CD8^+^ (**B**), CD20^+^ (**C**), and CD66b^+^ (**D**) cells under a power field of 200× magnification as well as corresponding areas of α-sma (**E**) and collagen (**F**) stained spots. CD3 is the marker of T cells, CD8—of cytotoxic T cells, CD20—of B cells, CD66b—of neutrophils, and α-sma—of cancer associated fibroblasts.

**Table 1 cancers-11-00039-t001:** Demographics of the study population.

Variables	Results
Age (Year, Mean ± SD)	70.44 ± 8.328
Gender (Female/Male)	31 (54.4%)/26 (45.6%)
Operation	23 PrPD (40.4%) 15 PPPD (26.3%) 13 Left resection (22.8%) 6 Pancreatectomy (10.5%)
Location	38 Head (66.7%) 7 Corpus (12.3%) 11 Tail (19.3%) 1 Corpus + tail (1.8%)
TNM Classification Grading	3 G1 (5.3%) 15 G2 (26.3%) 38 G3 (66.7%) 1 G4 (1.8%)
TNM Classification Tumor	1 T2 (1.8%) 55 T3 (96.5%) 1 T4 (1.8%)
TNM Classification Lymph Nodes	23 N0 (40.4%) 33 N1 (57.9%) 1 N2 (1.8%)
TNM Classification Resection	57 R0 (100%)
TNM Classification Metastasis	57 M0 (100%)
Tumor size (Centimetres)	1.4–8.0 cm (median 3.2 cm)
Necrosis	28 Grade 0 (49.1%) 19 Grade 1 (33.3%) 10 Grade 2 (17.5%)
CD3^+^ TILs (concentration)	0–39.667 (median 101)/0.145 mm^2^
CD8^+^ TILs (concentration)	0–72.467 (median 83)/0.145 mm^2^
CD20^+^ TILs (concentration)	0–752.333 (median 121)/0.145 mm^2^
CD66b^+^ TINs (concentration)	0–429.667 (median 59.333)/0.145 mm^2^
CAFs (Mean pixels)	0.390–46.752 (median 17.089)
Collagen (Mean pixels)	2.354–168.373 (median 56.37)

**Table 2 cancers-11-00039-t002:** Binary logistic regression of the different variables presented at time of resection. *p*-values < 0.05 were defined as statistically significant and appear in bold. * *p* < 0.001.

Variables	CD3	CD8	CD20	CD66b	α-sma	Collagen
Age	0.508	0.558	0.840	0.583	0.876	0.805
Gender	0.682	0.238	0.347	0.829	1.000	0.768
Operation	0.603	0.996	0.793	0.209	1.000	0.746
Location	0.316	0.483	0.597	0.748	0.561	0.711
TNM Grading	0.609	0.719	0.105	0.734	0.323	0.421
TNM Tumor	0.999	1.000	0.999	1.000	0.999	0.999
TNM Nodes	0.361	0.682	0.682	0.740	0.093	0.372
Tumor size	0.330	0.310	0.399	0.580	0.392	0.081
Necrosis	0.094	0.684	0.684	0.308	1.000	0.101
Thrombocytes	0.417	0.269	0.774	0.882	0.213	0.168
Albumin	0.362	0.301	0.735	0.467	0.799	0.078
CRP	0.796	0.673	0.079	0.870	0.877	0.220
Leukocytes	0.686	0.313	0.849	0.894	**0.029**	0.953
Haemoglobin	0.565	0.117	0.106	0.258	0.422	0.218
Creatinine	0.357	0.542	0.546	0.910	0.283	0.098
Bilirubin	0.194	0.303	0.889	0.819	0.699	0.136
γ-GT	0.433	0.694	0.854	0.479	0.924	0.576
AP	0.590	0.305	0.940	0.413	0.795	0.085
Lipase	0.404	0.804	0.717	0.923	0.997	0.601
CEA	0.073	0.543	0.229	0.356	0.733	0.444
CA 19-9	0.501	0.842	0.185	0.880	0.242	0.987
CD3		0.088	**0.049**	0.360	1.000	0.363
CD8	0.088		**0.003**	*	0.127	0.537
CD20	**0.049**	**0.003**		**0.003**	0.120	0.191
CD66b	0.360	*	**0.003**		**0.001**	0.516
α-sma	1.000	0.127	0.120	**0.001**		0.537
Collagen	0.363	0.537	0.191	0.516	0.537	

**Table 3 cancers-11-00039-t003:** Receiver Operating Characteristic (ROC) curves of high vs. low infiltration of immune cells and combined groups regarding the specificity and sensitivity to be alive after 24 months. Area under the curve (AUC). Significant results were bolted (*p* < 0.05). CD3 is the marker of T cells, CD8—of cytotoxic T cells, CD20—of B cells and CD66b—of neutrophils. Significant *p*-values were bolded.

TILs	AUC	95% CI	95% CI	*p*-Value
CD3	0.678	0.534	0.822	**0.025**
CD8	0.678	0.534	0.822	**0.025**
CD20	0.729	0.593	0.866	**0.004**
CD66b	0.587	0.434	0.740	0.272
CD3 + 8	0.687	0.539	0.835	**0.018**
CD3 + 20	0.687	0.539	0.835	**0.018**
CD3 + 66b	0.602	0.446	0.758	0.198
CD8 + 20	0.695	0.549	0.841	**0.014**
CD8 + 66b	0.621	0.469	0.774	0.125
CD20 + 66b	0.687	0.539	0.835	**0.018**
CD3 + 8 + 20	0.737	0.607	0.867	0.003
CD3 + 8 + 20 + 66b	0.693	0.554	0.832	**0.015**

**Table 4 cancers-11-00039-t004:** Univariate analysis in a linear regression of all independent variables on overall survival (OS) and progression-free survival (PFS) as dependent variables.

Variables	Overall Survival				Progression-Free Survival			
	β Coefficient	95% CI		*p*-Value	β-Coefficient	95% CI		*p*-Value
Age	0.000	−0.494	0.495	0.999	−0.053	−0.383	0.277	0.747
Gender	−4.736	−12.830	3.359	0.246	−3.351	−8.753	2.050	0.219
Operation	0.872	−4.301	3.659	0.872	0.342	−2.317	3.001	0.797
Location	−0.634	−5.384	4.116	0.790	−0.377	−3.552	2.798	0.813
TNM Grading	0.681	0.840	−6.045	0.406	−0.957	−5.446	3.531	0.671
TNM Tumor	−20.500	−41.575	0.575	0.056	−12.000	−26.197	2.197	0.096
TNM Nodes	1.000	−6.821	8.821	0.799	0.284	−4.945	5.513	0.914
Tumor size	−0.870	−4.119	2.378	0.593	−0.822	−2.987	1.344	0.450
Necrosis	−5.692	−10.890	−0.494	**0.032**	−3.638	−7.125	−0.152	**0.041**
CD3	−12.930	−20.309	−5.550	**0.001**	−8.555	−13.498	−3.613	**0.001**
CD8	−9.280	−17.050	−1.509	**0.020**	−5.256	−10.525	0.012	0.051
CD20	−8.340	−16.188	−0.492	**0.038**	−5.975	−11.188	−0.763	**0.025**
CD66b	−3.395	−11.570	4.780	0.409	−1.344	−6.829	4.142	0.625
α-sma	4.810	−5.050	14.669	0.330	1.095	−4.707	6.897	0.705
Collagen	−11.427	−20.726	−2.128	**0.017**	−6.864	−12.254	−1.473	**0.014**
Stroma index	−3.988	−8.266	0.290	0.067	−1.755	−4.294	0.784	0.170
Distance CD8	3.576	−6.191	13.342	0.464	−0.164	−6.791	6.464	0.961
Distance CD20	−6.612	−15.404	2.180	0.137	0.406	−5.649	6.460	0.893
Thrombocytes	0.02	−0.044	0.048	0.931	0.013	−0.017	0.044	0.383
Albumin	−2.121	−8.740	4.498	0.517	0.551	−4.426	5.528	0.822
CRP	−0.718	−2.121	0.685	0.309	−0.343	−1.263	0.576	0.457
Leukocytes	0.000	−0.001	0.002	0.847	0.000	−0.001	0.001	0.786
Haemoglobin	0.373	−1.810	2.557	0.733	0.516	−0.938	1.971	0.480
Creatinine	−1.790	−12.366	8.785	0.736	−2.448	−9.492	4.596	0.489
Bilirubin	0.523	−0.240	1.286	0.175	0.276	−0.237	0.789	0.285
γ-GT	0.001	−0.005	0.008	0.730	0.002	−0.002	0.006	0.341
AP	0.007	−0.009	0.024	0.376	0.002	−0.009	0.013	0.700
Lipase	−0.058	−0.174	0.057	0.309	−0.019	−0.096	0.057	0.610
CEA	−0.037	−0.071	−0.003	**0.033**	−0.016	−0.039	0.008	0.181
CA 19-9	−0.329	−0.972	0.314	0.309	−0.297	−0.729	0.134	0.172
Adjuvant	0.006	−0.002	0.014	0.132	0.017	0.006	0.028	**0.004**
CD3/α-sma	0.282	−0.004	0.567	0.053	0.205	−0.292	0.701	0.410
CD3/collagen	−0.030	−0.402	0.343	0.873	0.123	−0.499	0.746	0.691
CD8/α-sma	−0.207	−3.465	3.051	0.899	0.691	−4.899	6.282	0.804
CD8/collagen	−0.090	−0.271	0.092	0.324	−0.112	−0.425	0.201	0.475
CD20/α-sma	0.484	−0.561	1.529	0.355	0.438	−1.370	2.246	0.627
CD20/collagen	−0.107	−0.501	0.287	0.586	−0.034	−0.713	0.644	0.919
CD66b/α-sma	−0.599	−2.122	0.925	0.432	−0.526	−3.059	2.007	0.678
CD66b/collagen	−0.174	−0.323	−0.024	**0.024**	−0.263	−0.512	−0.014	**0.039**

*p*-values under 0.050 appear in bold (*n* = 57). The variable “adjuvant” describes whether the patient received a completed adjuvant chemotherapy with gemcitabine. The variables with the tumor-infiltrating leukocytes (TILs and TINs) describe groups of high or low infiltration in relation to the median value. Significant *p*-values were bolded.

**Table 5 cancers-11-00039-t005:** Multivariate Analysis of Variables with *p* < 0.05 in the univariate analysis as a linear regression. Combination groups of different immune cell types were excluded due to possible statistical effects by double-use. *p*-values under 0.050 appear in bold (*n* = 57). Significant *p*-values were bolded.

Variables	Overall Survival	Progression-Free Survival
β Coefficient	95% CI	*p*-Value	β-Coefficient	95% CI	*p*-Value
CD3	−9.477	−18.012	−0.943	**0.031**	−6.250	−11.413	−1.087	**0.019**
CD8	−8.643	−17.439	0.152	0.054				
CD20	−2.756	−11.713	6.201	0.537	−2.165	−7.477	3.147	0.415
Collagen	−10.763	−19.181	−2.346	**0.014**	−6.416	−11.669	−1.163	**0.018**
CEA	−0.024	−0.059	0.011	0.166

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
