# Peer review of "Prognostic Impact of Tumor-Infiltrating Lymphocytes and Neutrophils on Survival of Patients with Upfront Resection of Pancreatic Cancer"

_cancers, 2019, doi:10.3390/cancers11010039_

Reviewer 1 Report

Dr Miksch and colleagues describe the importance of peritumoral TILs and immune subtype classification in PDAC. Their results illustrated that high densities of peritumoral hot spots with CD3+, CD8+, and CD20+ TILs correlated significantly with improved OS and PFS. Combined immune cell subtypes predicted improved OS and PFS. High infiltration of CD3+ TILs predicted progression after 12 months. The location of TILs hot spots and their distance to the tumor front did not correlate with patient survival. Peritumoral TILs and the composition of the stroma predict OS and PFS in PDAC.

However, it is difficult to understand how Accumulation of peritumoral TILs in PDAC correlates with better survival since patients underwent surgery, and therefore, TILs and the tumor mass are not longer there. Is there any information of the PDACs when patients pass away? (Histology, grade, TILs, metastasis,…)

It is not clear for the survival patients that were treated or not: Sixteen patients (28.1%) completed adjuvant chemotherapy with gemcitabine.

Authors should include some other information about the tumor characteristics (tumor size, presence of necrosis, grade of differentiation, ratio of tumor cells/stroma, vessel density, other markers for fibroblasts such as vimentin,…).

It could be important also to know the presence of immune checkpoints (ie: PD1) as well as GATA6. Please, include GATA6 expression (marker for the Classic tumor subtype) in the studies for survival.

Recommendation for figures 2 and 3: Maintain the green color for the low infiltration.

Please, indicate the n of patients in all the curves.

Finally, please revise the English; there are sentences with typos.

Example: However, therapeutic approaches fosucing on targets definded by genomics may only relevant for a small proportion of patients with PDAC

Author Response

Reviewer 1

Dr Miksch and colleagues describe the importance of peritumoral TILs and immune subtype classification in PDAC. Their results illustrated that high densities of peritumoral hot spots with CD3+, CD8+, and CD20+ TILs correlated significantly with improved OS and PFS. Combined immune cell subtypes predicted improved OS and PFS. High infiltration of CD3+ TILs predicted progression after 12 months. The location of TILs hot spots and their distance to the tumor front did not correlate with patient survival. Peritumoral TILs and the composition of the stroma predict OS and PFS in PDAC.

We would like to thank the Reviewer 1 for deeping understanding ouf our work and helpful comments to improve our work.

However, it is difficult to understand how Accumulation of peritumoral TILs in PDAC correlates with better survival since patients underwent surgery, and therefore, TILs and the tumor mass are not longer there. Is there any information of the PDACs when patients pass away? (Histology, grade, TILs, metastasis,…)

The accumulation of peritumoral TILs and TINs was related with survival in our study. The subtype classification of patients with PDAC may be feasible by including quantification of the immune cell infiltration in resected patients to the TNM classification. A multicentre study about the immune cell infiltration in patients with resected colorectal cancer supported the implementation of the Immunoscore as a new component of a TNM-Immune classification of cancer [1]. Subtype classification of PDAC by histologic signature could be an important tool to predict prognosis more precisly and select patients for adjuvant therapy [2, 3]. Furthermore, tumor-infiltrating leukocytes are reflecting the immune response. This study shows the accumulation in a paraffin tissue at the time of resection.

At time of resection effector and memory immune cells are part of the immune respons which was observed and quantified in this study with the peritumoral hot spots of infiltrating leukocytes. In cases of immunogenetic reactivation the existing memory cells may inhibit the tumor dissemination [4]. Therefore, a positive immune infiltration related with improved survival may reflect a positive immune response during follow-up.

Information of patients’ TNM classification [5] after upfront resection were as followed (n=57) and can be found in Table 1 in the manuscript:

pT stage pT1=0 (0%), pT2=1 (1.8%), pT3=55 (96.5%), and pT4=1 (1.8%); lymph node stage pN0=23 (40.4%), pN1=33 (57.9%), and pN2=1 (1.8%); metastasis pM0=57 (100%); grading pG1=3 (5.3%), residual tumor pR0=57 (100%).

There is no histological data availabe at time of death of the patients in this study. We did not perform any additional surgery after primary resection in this study population. Nevertheless, we found that 56 patients of this study had tumor recurrence or metastasis during follow-up (median follow-up of 19.0 months).

It is not clear for the survival patients that were treated or not: Sixteen patients (28.1%) completed adjuvant chemotherapy with gemcitabine.

16 patients (28.1%) completed adjuvant chemotherapy with gemcitabine, whereas 41 patients (71.9%) did not complete gemcitabine treatment due to death, progression or complications during chemo therapy. Therefore, the statistical impact of completed adjuvant chemotherapy with gemcitabine has to be discussed critically: It is important to note that gemcitabine was the only kind of adjuvant therapy in this study population. Completed adjuvant chemotherapy with gemcitabine did no correlate significantly with overall survival (p=0.132) but with progression-free survival (p=0.004). This is described in the chapter 2.1. Demographical data and the statistical results in Table 4.

Authors should include some other information about the tumor characteristics (tumor size, presence of necrosis, grade of differentiation, ratio of tumor cells/stroma, vessel density, other markers for fibroblasts such as vimentin,…).

We reexamined our pancreatic cancer tissue and added characteristics about size and presence of necrosis to manuscript.

The diameter of the tumor (range 1.4 – 8.0 cm (median 3.2 cm)) did not correlate with survival. Furthermore, there was no significant correlation with the infiltration of lymphocytes, neurophils, and the densitiy of fibroblasts but a trend for high density of collagen (p=0.081). The variable tumor size was added to Table 1, 2, and 4 in the manuscript.

The grades of differentiation were pG2=15 (26.3%), pG3=38 (66.7%), and pG4=1 (1.8%) and did not correlate with survival (p=0.406 for OS, p=0.671 for PFS).

We included the ratio immune cells/stroma to our manuscript as a further tool to describe a subtype classification of patients with PDAC. The mean number of infiltrated cells and mean area of stromal composition type were selected for univariate linear regression (Table 4 in the manuscript):

Variables

Overall Survival

Progression-Free Survival

ß Coefficient

95% CI

p-Value

ß-Coefficient

95% CI

p-Value

CD3/α-sma

0.282

-0.004

0.567

0.053

0.205

-0.292

0.701

0.410

CD3/collagen

-0.030

-0.402

0.343

0.873

0.123

-0.499

0.746

0.691

CD8/α-sma

-0.207

-3.465

3.051

0.899

0.691

-4.899

6.282

0.804

CD8/collagen

-0.090

-0.271

0.092

0.324

-0.112

-0.425

0.201

0.475

CD20/α-sma

0.484

-0.561

1.529

0.355

0.438

-1.370

2.246

0.627

CD20/collagen

-0.107

-0.501

0.287

0.586

-0.034

-0.713

0.644

0.919

CD66b/α-sma

-0.599

-2.122

0.925

0.432

-0.526

-3.059

2.007

0.678

CD66b/collagen

-0.174

-0.323

-0.024

0.024

-0.263

-0.512

-0.014

0.039

As authors, we are aware that there will always be a lack of complete information dealing with the tumor microenvironment. The amount and kind of antibodies were selected to reflect leukocyte infiltration. With this we set out to discover a subtype classification based on the proposed four different immune cell types. However, further research is needed to gain a more complete picture of the tumor immune stroma and its effects on PDAC patients who could be resected. Based on our results we have intiated further studies to elucidate specific aspects of this topic.

Rating of tumor necrosis was performed as recommended. The extend of necrosis within the tumor tissue was assessed by a board certified pathologist (SO) on HE stained slide of the same sample used in the immunhistochemical stainings. Briefly, absent or insignificant necrosis of single cells was graded as score 0, spot like necrosis in limited zones within the tissue as score 1, and extensive or confluent necrotic areas within the tumor tissue as score 2. Finally, the necrosis rating was correlated with survival and results added to the manuscript (Supp. Figure 1):

Presence of extensive necrosis was related with impaired overall survival (p<0.001) and progression-free survival (p=0.009). Bigger tumor size was related with extensive grade of necrosis in the regression analysis (p=0,000526).

Furthermore, grade of necrosis did not correlate with immune cell infiltration (Table 2) or the serum levels of leukocytes/CRP.

It could be important also to know the presence of immune checkpoints (ie: PD1) as well as GATA6. Please, include GATA6 expression (marker for the Classic tumor subtype) in the studies for survival.

The aim of this study was to investigate the revelance of the quantification of the tumor immune stroma (QTiS) and therefore did not include other cell types [6]. So far, the importance of peritumoral immune cell accumulation was highlighted and high infiltration of CD3+, CD8+, and CD20+ TILs were related with improved survival after upfront resection in PDAC. Furthermore, the QTiS algorithm showed significant results for immune cell infiltration in patients with hepatocellular carcinoma [7].

Sure, quantifying the tumor NON-immune microenvironment can be important. Concerning PD1-PDL1 axe, it is known that it is upregulated in PDAC and related to impaired survival [8]. Regarding GATA6, we are carrying out a project about GATA6 expression in PDAC tissues in frame of huge multicentral project. The analysis of this project will be done next year.

Recommendation for figures 2 and 3: Maintain the green color for the low infiltration.

We changed the color according to the reviewer’s comment.

Please, indicate the n of patients in all the curves.

We changed the figures according to the reviewers comment (Fig. 1-4 and Supp. Figures).

Finally, please revise the English; there are sentences with typos.

Example: However, therapeutic approaches fosucing on targets definded by genomics may only relevant for a small proportion of patients with PDAC

We improved the sentence mentioned in the example and revised the whole manuscript concerning English language.

References

1.         Pages, F., et al., International validation of the consensus Immunoscore for the classification of colon cancer: a prognostic and accuracy study. Lancet, 2018. 391(10135): p. 2128-2139.

2.         Mahajan, U.M., et al., Immune Cell and Stromal Signature Associated with Progression-free Survival of Patients with Resected Pancreatic Ductal Adenocarcinoma. Gastroenterology, 2018.

3.         Hilmi, M., L. Bartholin, and C. Neuzillet, Immune therapies in pancreatic ductal adenocarcinoma: Where are we now? World J Gastroenterol, 2018. 24(20): p. 2137-2151.

4.         Beckhove, P., et al., Specifically activated memory T cell subsets from cancer patients recognize and reject xenotransplanted autologous tumors. J Clin Invest, 2004. 114(1): p. 67-76.

5.         Kamarajah, S.K., et al., Validation of the American Joint Commission on Cancer (AJCC) 8th Edition Staging System for Patients with Pancreatic Adenocarcinoma: A Surveillance, Epidemiology and End Results (SEER) Analysis. Ann Surg Oncol, 2017. 24(7): p. 2023-2030.

6.         Miksch, R.C., et al., Development of a reliable and accurate algorithm to quantify the tumor immune stroma (QTiS) across tumor types. Oncotarget, 2017. 8(70): p. 114935-114944.

7.         Schoenberg, M., Hao J, Bucher JN, Miksch RC, Anger HJW, Mayer B, Mayerle J, Neumann J, Guba MO, Werner J, Bazhin AV, Perivascular Tumor-Infiltrating Leukocyte Scoring for Prognosis of Resected Hepatocellular Carcinoma Patients. Cancers, 2018. 10(10).

8.         Loos, M., et al., Clinical significance and regulation of the costimulatory molecule B7-H1 in pancreatic cancer. Cancer Lett, 2008. 268(1): p. 98-109.

Reviewer 2 Report

The authors provide an immunohistochemistry analysis of  of 57 resected treatment naive pancreatic cancer patients and correlate tumor infiltrating lymphocytes with PFS and OS. As the authors do not provide analysis on MDSC, it would be best to mention the limitation of the correlation without providing analysis on the presence of MDSC. 

minor comments: 

line 121 indicates that 3 cell types are analyzed in  both panel G and H in Figure 3 but panel H has 4 cell types indicated next to the graph. 

Author Response

Reviewer 2

The authors provide an immunohistochemistry analysis of  of 57 resected treatment naive pancreatic cancer patients and correlate tumor infiltrating lymphocytes with PFS and OS. As the authors do not provide analysis on MDSC, it would be best to mention the limitation of the correlation without providing analysis on the presence of MDSC.

We appreciate your very important comments and improved the design and our manuscript.

The immunosuppression in PDAC within the tumor microenvironment is thought to impair the host's antitumor response by multiple effector cells like myeloid-derived suppressor cells (MDSC) [1]. MDSCs can suppress T cell effector cells and induce immune tolerance for immune evasion [2]. A complex system of T cells, pancreatic tumor cells, myeloid-derived suppressor cells, tumor-associated macrophages, regulatory T cells (Tregs), and pancreatic stellate cells (PSCs) seem to interact. There is still more evidence and research needed to totally understand the complex tumor microenvironment [3]. Our results suggest that tumor-infiltrating leukocytes are related with survival after resection. Hence, this study has its limitations because of the complexity of the tumor microenvironment which should be regarded if conclusions had been drawn. We added this part to the final lines of our discussion in the manuscript.

minor comments:

line 121 indicates that 3 cell types are analyzed in  both panel G and H in Figure 3 but panel H has 4 cell types indicated next to the graph.

We changed the sentences and added a separate sentence to describe panel H in Figure 2 and 3.

References

1.         Pergamo, M. and G. Miller, Myeloid-derived suppressor cells and their role in pancreatic cancer. Cancer Gene Ther, 2017. 24(3): p. 100-105.

2.         Ren, B., et al., Tumor microenvironment participates in metastasis of pancreatic cancer. Mol Cancer, 2018. 17(1): p. 108.

3.         Habtezion, A., M. Edderkaoui, and S.J. Pandol, Macrophages and pancreatic ductal adenocarcinoma. Cancer Lett, 2016. 381(1): p. 211-6.

Reviewer 3 Report

The paper is about Tumor infiltrating leukocytes but the authors refer to these as TILs. In the oncology literature TILs are always tumor infiltrating lymphocytes, Then they go on to do flow and do CD3, CD8 etc which are lymphocyte markers and they refer to these also as TILs. 

I cannot tell by the paper if they are referring to neutrophils or lymphocytes.

Very confusing.

Most consider differentiating TILs (lymphocytes ) from leukocytes by calling the later neutrophils so maybe they should differentiate by calling the tumor infiltrating neutrophils TINs 

Author Response

Reviewer 3

The paper is about Tumor infiltrating leukocytes but the authors refer to these as TILs. In the oncology literature TILs are always tumor infiltrating lymphocytes, Then they go on to do flow and do CD3, CD8 etc which are lymphocyte markers and they refer to these also as TILs.

I cannot tell by the paper if they are referring to neutrophils or lymphocytes.

Very confusing.

Most consider differentiating TILs (lymphocytes ) from leukocytes by calling the later neutrophils so maybe they should differentiate by calling the tumor infiltrating neutrophils TINs

Dear Reviewer, thanks a lot for your helpful comments. We improved the descriptions and manuscript as recommended.

In this study we describe the importance of tumor-infiltrating leukocytes. T cells (CD3, CD8), B cells (CD20), and Neutrophils (CD66b) are part of the leukocyte cell group. According to reviewer’suggestion we changed and refer TILs as tumor-infiltrating lymphocytes (CD3, CD8, and CD20), and TINs as tumor-infiltrating neutrophils. TILs and TINs are part of the group of tumor-infiltrating leukocytes.

Round  2

Reviewer 1 Report

This study is now ready for publication

Author Response

Dear reviewers,

We are very happy about your revisions and thank you very much for your valuable comments. In the manuscript we used the “track changes” mode in word to highlight any revisions that were done in the text of the manuscript.

Reviewer 1

This study is now ready for publication.

Dear reviewer, we thank you very much for your support and thanks a lot for your valuable comments to improve our manuscript.

Reviewer 3 Report

Much improved and understandable now. Only some very minor suggestions:

71 mechanisms helping PDAC cells to escape the anti-tumor immune response consisting of tumor-

72 infiltrating leukocytes, especially TILs (including regulatory T cells), TINs, regulatory T cells, M2 macrophages, neutrophils,and

73 myeloid-derived suppressor cells [35]

Can the authors possibly add to their METHODS and /Or to Table 3 for non-immunology readers what each antibody is specific for: ie. CD3 lymphocytes, CD66 (neutrophils)

Author Response

Dear reviewers,

We are very happy about your revisions and thank you very much for your valuable comments. In the manuscript we used the “track changes” mode in word to highlight any revisions that were done in the text of the manuscript.

Reviewer 3

Much improved and understandable now.

Only some very minor suggestions:

71 mechanisms helping PDAC cells to escape the anti-tumor immune response consisting of tumor-

72 infiltrating leukocytes, especially TILs (including regulatory T cells), TINs, regulatory T cells, M2 macrophages, neutrophils, and

73 myeloid-derived suppressor cells [35]

Can the authors possibly add to their METHODS and /Or to Table 3 for non-immunology readers what each antibody is specific for: ie. CD3 lymphocytes, CD66 (neutrophils)

Dear Reviewer, thanks a lot for your helpful comments and assistance. We improved the descriptions and manuscript as recommended.

We deleted “regulatory T cell” and “neutrophils” in line 72 as suggested. Furthermore, we added a description of the different antigens shown in the manuscript: CD3 is the marker of T cells, CD8 is one of cytotoxic T cells, CD20 - of B cells, and CD66b - of neutrophils. This information was added to Table 3, Figure 5, and the section “4.3 Immunohistochemistry”.